Age, growth, mortality and recruitment of thin sharpbelly Toxabramis swinhonis Günther, 1873 in three shallow lakes along the middle and lower reaches of the Yangtze River basin, China

Dong Xianghong 1 2 3
Xiang Tao 1 2
Ju Tao 4
Li Ruojing 1 2
Ye Shaowen 1
Lek Sovan 3
Liu Jiashou 1 jsliu@ihb.ac.cn
Grenouillet Gaël 3
1 State Key Laboratory of Freshwater Ecology and Biotechnology, Institute of Hydrobiology, Chinese Academy of Sciences , Wuhan , China
2 University of Chinese Academy of Sciences , Beijing , China
3 Laboratoire Evolution & Diversité Biologique, Université Toulouse III , Toulouse , France
4 Key Laboratory of Freshwater Biodiversity Conservation, Ministry of Agriculture of China, Yangtze River Fisheries Research Institute, Chinese Academy of Fishery Science , Wuhan , China
Chavanich Suchana
Electronic publication date: 2019 Apr 12
Publication date: 2019
Volume: 7
Electronic Location ID: e6772
Received 2018 Sep 7; Accepted 2019 Mar 12
Copyright: © 2019 Dong et al.
Copyright year: 2019
Copyright holder: Dong et al.
License: This is an open access article distributed under the terms of the Creative Commons Attribution License, which permits unrestricted use, distribution, reproduction and adaptation in any medium and for any purpose provided that it is properly attributed. For attribution, the original author(s), title, publication source (PeerJ) and either DOI or URL of the article must be cited.
License URL: https://creativecommons.org/licenses/by/4.0/

Keywords: Toxabramis swinhonis, Age and growth, Mortality and recruitment, The Yangtze River basin, Shallow lakes

Funding: National Science and Technology Supporting Program of China 2012BAD25B08 Earmarked Fund for China Agriculture Research System CARS-45 State Key Laboratory of Freshwater Ecology and Biotechnology 2014FBZ04 China Scholarship Council (CSC) This work was supported by the National Science and Technology Supporting Program of China (2012BAD25B08), the Earmarked Fund for China Agriculture Research System (CARS-45), the State Key Laboratory of Freshwater Ecology and Biotechnology (2014FBZ04), and the China Scholarship Council (CSC). The funders had no role in study design, data collection and analysis, decision to publish, or preparation of the manuscript.

==============================
Despite being the most dominant and widespread small fish species in the lakes along the middle and lower reaches of the Yangtze River basin, Toxabramis swinhonis has been paid little attention by fisheries scientists and little is known about its population characteristics. For this reason, we estimated age, growth, mortality and recruitment of this species based on three shallow lakes, Biandantang Lake, Shengjin Lake and Kuilei Lake (BDT, SJH and KLH, respectively) in this region. A total of 13,585 (8,818 in BDT, 2,207 in SJH and 2,560 in KLH) individuals were collected during monthly sampling from July 2016 to September 2017. The results revealed that the age structures of T. swinhonis consisted of four age groups (0+–3+), with 0+–1+ year old fish comprising more than 98% of the samples. Allometric growth patterns were displayed by fish from all sampling sites and the von Bertalanffy growth functions estimated were Lt = 173.25 (1 – e−1.20 (t + 1.09)): BDT; Lt = 162.75 (1 – e−1.20 (t + 1.08)): SJH and Lt = 215.25 (1 – e−1.20 (t + 1.12)): KLH, respectively. The rates of total mortality (Z), natural mortality (M) and fishing mortality (F) at BDT, SJH and KLH were computed as 5.82, 5.50 and 4.55 year−1; 1.89, 1.87 and 1.75 year−1; 3.93, 3.63 and 2.80 year−1, respectively. Meanwhile, growth performance indices (φ′) were 0.68 (in BDT), 0.66 (in SJH) and 0.62 (in KLH), which indicated that T. swinhonis were overfished slightly in all study areas. Area-specific recruitment patterns were similar to each other, displaying evidence of batch spawning, with major peaks in April and August, accounting for 92.21% (BDT), 88.21% (SJH) and 88.73% (KLH) of total recruitment, respectively. These results showed that brief generation-time, fast growth rate, relatively high natural mortality rate and strong reproductive capacity (r-strategies) are reasons why this species became the most dominant species in many lakes of China.

Introduction

Many lakes (about 4,000) distributed along the middle and lower reaches of the Yangtze River basin (China) account for approximately 1/3 of the total area of lakes in China (Liu & He, 1992; Li, 2004). These lakes are typically shallow (no thermal stratification) with numerous submerged aquatic macrophytes and generally have higher fisheries productivity (Li, Zhang & Li, 2010; Ye et al., 2014). However, due to intensive removal of piscivores and overfishing by commercial fisheries in the past decades, short-lived small fish such as Hyporhamphus intermedius (Cantor, 1842), Hemiculter leucisculus (Basilewsky, 1855), Coilia ectenes taihuensis (Yen et Lin, 1976), Cultrichthys erythropterus (Basilewsky, 1855), Pseudobrama simoni (Bleeker, 1864) and Toxabramis swinhonis Günther, 1873 gradually become dominant species in these lakes and account for about 2/3 of total species (40–70 species in one lake) (Li et al., 1988; Cao et al., 1991; Liu et al., 2005; Mao et al., 2011; Ye et al., 2014). T. swinhonis was the most dominant of these species, accounting for more than 85% of total individuals caught in some lakes (Ge, Zhong & Tang, 2009) and showing an increase trend in abundance (Li et al., 2018), which elevated our concerns on the sustainability of fisheries resources in these water bodies.

Besides, despite not a fishery target, T. Swinhonis is still very valuable in terms of ecology and fisheries management as it is one of the main bycatch species in China’s freshwater fishery, an important potential food resource for the carnivorous aquatic organisms (Dai et al., 2015; Ye et al., 2014) and plays a key role in the energy flows of their living hydro-ecosystems, linking zooplankton (as T. swinhonis mainly feeds upon copepods, cladocerans and insect larvaes) to higher trophic levels (Zhang, 2005). Meanwhile, many fisheries scientists pointed out that there existed a close relationship between population expansion and particular features of biology and ecology (Peter, 1980; Wang et al., 2013; Guo et al., 2016). Thus, it is the time that much more attentions should be paid on the biology and ecology of this outbreak species.

Published information on the biology and ecology of T. swinhonis is scarce. Li et al. (1988) defined this species as a small coarse fish and speculated on population control methods based on two reservoirs in northern China. Xie, Cui & Li (2001a, 2001b) investigated the relationship between diet and morphology of this species and also examined the association of submerged macrophytes in Liangzi Lake, China to density and biomass of this species. Ye et al. (2014) studied the growth and mortality of this species using length-based approaches in Niushan Lake, China.

In order to better understand the biology and ecology of this species and explain why it has become the most dominant fish species in many freshwater lakes of China, we conducted the present study to: (a) provide novel and comprehensive population characteristics of T. swinhonis, especially age structure (first report in the study regions), (b) identify demographic characteristics consistent with rapid population expansion and (c) provide potential strategies for fishery management on T. swinhonis.

Materials and methods

Study sites

The present study was performed in three shallow lakes (no thermal stratification), Biandantang Lake (BDT) (114°43′E, 30°15′N), Shengjin Lake (SJH) (117°03′E, 30°23′N) and Kuilei Lake (KLH) (120°51′E, 31°24′N), along the middle and lower reaches of the Yangtze River basin, China. These lakes are scattered longitudinally in this region. BDT, SJH and KLH are upstream, midstream and downstream, respectively (Fig. 1).

Figure 1 Map of the study area and locations of the three sampling sites for T. swinhonis (in the middle and lower reaches of Yangtze River basin, China).

BDT, SJH and KLH are the abbreviations for Biandantang Lake, Shengjin Lake and Kuilei Lake, respectively.

Data collection and fish sampling

Toxabramis swinhonis were collected monthly from July 2016 to September 2017 using multi-mesh gillnets (1.5 × 30 m) (Appelberg et al., 1995). In order to obtain sufficient samples, sampling time in each lake was at least 3 days monthly. In addition, larval and juvenile T. swinhonis were caught by fishermen using electrofishing gear to ensure a representative and unbiased assessment of their biological characteristics. Total lengths (L) were measured to nearest 0.01 mm with a vernier caliper and body weights (W) were recorded to the nearest 0.01 g with a precision balance in the field. Sex was determined macroscopically. Scales were removed randomly from the middle portion of the lateral body of 30 individuals per month from each site and stored in small envelopes labeled with sampling site and morphometric data (Narejo et al., 2009; Zhang & Li, 2002). Lake surface temperatures (LST) were recorded every 4 h throughout the study period using HOBO Temp/Light data loggers (start time 8 am) (da Cunha, 2015; Dalton et al., 2016). All the procedures described here were approved by the ethics committee of the Institute of Hydrobiology, Chinese Academy of Sciences (Y216011101).

Length distribution and length–weight relationship

Kolmogorov–Smirnov tests (K–S tests) were used to evaluate the similarity of length–frequency distribution between different sampling sites and normality of data. Differences in mean L of T. swinhonis among sites were evaluated using Kruskal–Wallis test with Tukey’s post-hoc when the error or data was not a normal distribution. The relationship between length and weight was calculated for each site using power regression equation:W=aLbeε,ε∼N(0,σ2),

where a is the intercept of the regression or shape coefficient and b is the allometric or slope parameter (Ricker, 1975). The optimal regression parameters were gained by minimizing the residuals errors using the ordinary least square method (O’Brien, 2012). Student’s t-test was utilized to test whether the slope of regression was significantly different from 3, indicating the growth pattern of fish: isometric (b = 3, no change of density and shape as one fish grows), positive allometric (b > 3, the fish becomes relatively stouter or deeper-bodies as it becomes longer) or negative allometric (b < 3, fish becomes slimmer as it grows) (Ye et al., 2007). Generally, the allometric coefficient (b) is within the range from 2 to 4 for most fish species (Koutrakis & Tsikliras, 2003), and thus this thumb rule can be used to corroborate the validation of the length and weight relationships of T. swinhonis in our study. All statistical analyses were performed using R software version 3.3.2 (R Core Team, 2017) with significance level (α) equal to 0.05.

Growth

Growth of a fish can be described as an increase in either length or weight. Several mathematical equations have been created to model fish growth. One of these, the von Bertalanffy growth model (VBGM), is widely used to model growth of fishes. Thus, we used the VBGM to describe the growth of T. swinhonis in the current study. The typical VBGM is represented by (Von Bertalanffy, 1938):Lt=L∞(1−e−K(t−t0)),

where Lt is the expected or average L at time (or age) t, L∞ is the mean asymptotic L, K is the rate at which L∞ is approached and t0 is the hypothetical age at which fish L equals zero. While VBGM parameters can be estimated by length (weight)-age approach, the length-based method is most extensively adopted, especially in the tropical and subtropical regions (Sparre & Venema, 1998). In this study, length–frequency datasets with a constant class size (5 mm) were used to obtain the optimal growth parameters (corresponding to the maximum of the goodness of fit index, Rn) using the FiSAT II software (Gayanilo, Sparre & Pauly, 2005) and the module of ELEFAN I (Pauly, 1986). The theoretical age (t0) was determined by the empirical equation of Pauly (1983):log10(−t0)=−0.392−0.275 log10(L∞)−log10(K).

In order to compare the growth performance of different geographical populations, the growth performance index (GPI) φ′ (phi-prime) was calculated using the following formula (Pauly & Munro, 1984):ϕ′=log10(K)+2log10(L∞).

Additionally, in order to verify the accuracy of length-based analysis, the potential or expected longevity (tmax) of T. swinhonis was calculated based on Pauly’s (1983) empirical equation:tmax=3K,

where tmax is the approximate maximum age of T. swinhonis at each study site, and K is the growth constant in von Bertalanffy growth function.

Age estimate

Scales of T. swinhonis were immersed in 10% NaOH solution for 3 h, cleaned with running water and examined under a dissecting microscope to determine age (Steinmetz & Müller, 1991). Four to six scales were examined from each fish, and age was eventually defined as the most frequently occurring age from the scales.

Mortality coefficients and fisheries status

To obtain the total mortality rates (Z), the length-converted catch curves were applied to the pooled length frequency datasets using the estimated growth parameters. This process was done using the FiSAT II package and mortality menu (Gayanilo, Sparre & Pauly, 2005).

The natural mortality rates (M) were obtained from Pauly’s (1980) empirical equation:log10M=−0.0066−0.279log10L∞+0.6543log10K+0.4634log10T,

where T is the average annual LST (°C) (in the present study, along the direction of increase of longitude, T = 17.5, 16.6 and 17.1 °C, respectively), a proxy for the habitat temperature (Pauly, 1980). The validity of estimates of M can be judged by the M/K ratio as this ratio has been demonstrated to be within the range of 1.12–2.50 for most species around the world (Beverton & Holt, 1957).

Fishing mortality (F) was estimated from the equation F = Z – M (Gulland, 1965).

We estimated exploitation rate (E) using the equation of Elliott (1983):E=FZ.

Recruitment time and pattern

Timing and patterns of recruitment to T. swinhonis stocks were determined by backward projection of the length–frequency datasets onto the time axis of a time-series of samples as described in FiSAT II (Moreau & Cuende, 1991). This routine can provide the relative spawning time and the number of recruitment pulses (recruitment pattern) per year.

Results

Length–frequency distribution and age structure

A total of 13,585 individual T. swinhonis were measured during 15 months of field sampling. The length–frequency distributions of T. swinhonis are shown in Fig. 2. Lengths of T. swinhonis ranged from 53.32 to 165.16 mm L (mostly between 60 and 140 mm) in BDT, from 51.39 to 154.35 mm (mostly between 80 and 140 mm) in SJH and from 61.94 to 198.85 mm (mostly between 100 and 180 mm) in KLH. Results from K–S tests indicated that length frequency distribution data were non-normal in three lakes (for all three K–S tests, D = 1, P ≈ 0 (< 2.2 × 10−16), E = 1). Individual comparisons of these distributions indicated statistical differences between lakes (for BDT and SJH, D = 0.43, P ≈ 0 (< 2.2 × 10−16), E = 0.43; for BDT and KLH, D = 0.75, P ≈ 0 (< 2.2 × 10−16), E = 0.75 and for SJH and KLH, D = 0.56, P ≈ 0 (< 2.2 × 10−16), E = 0.56). Kruskal–Wallis test (χ2 = 5,186.90, df = 2, P ≈ 0 (< 2.2 × 10−16), E = 0.28) revealed that there was a statistical difference in mean L among different sampling sites and represented a trend of L increasing with the increasing longitude (Fig. 3).

Figure 2 Annual total length frequency distribution of T. swinhonis in (A) BDT, (B) SJH and (C) KLH.

Figure 3 Boxplot (horizontal line within box: median; boundaries of the box: first and third quartiles; bars: lower and upper inner fences; circles: outliers) about Kruskal–Wallis test in the total length (L) of T. swinhonis at different sampling sites.

Age structure of the T. swinhonis specimens collected from the three lakes, as determined from scales, were simple and consisted of four age groups ranging from 0+ to 3+ years (Fig. 4), similar to results obtained using Pauly’s (1983) empirical equation for maximum age (tmax = 3/K, all sites were 2.5). The dominant age group was 0+–1+ year old fish at all three sites (98%) (Table 1). These data suggest that T. swinhonis is a short-lived fish species. A chi-square test found significant differences among the age structures at the three sampling sites (χ2 = 103.82, df = 6, P = 4.00 × 10−20, E = 0.18).

Figure 4 The scales of T. swinhonis from BDT Lake with (A), (B), (C) and (D) corresponded with 0+, 1+, 2+ and 3+ years, respectively.

Blue dots represented the annuli.

Table 1 Mean observed length (mm)-at-age for T. swinhonis fish from Biandantang Lake, Shengjin Lake, and Kuilei Lake.

Age	Biandantang Lake	Shengjin Lake	Kuilei Lake	
Mean ± S.E. (mm)	n	Percentage (%)	Mean ± S.E. (mm)	n	Percentage (%)	Mean ± S.E. (mm)	n	Percentage (%)	
1	102.99 ± 1.01	307	74.15	113.85 ± 0.87	316	65.83	131.19 ± 1.09	298	45.43	
2	111.05 ± 1.28	99	23.92	127.97 ± 0.79	157	32.71	153.73 ± 0.74	351	53.51	
3	116.91 ± 7.16	6	1.45	131.46 ± 2.62	7	1.46	166.60 ± 8.24	7	1.06	
4	131.91 ± 13.68	2	0.48	NA	NA	0.00	NA	NA	0.00	
Note:

NA, not available; n, the number of specimen; S.E., standard error.

The length–weight relationships and growth pattern

Length and weight relationships for each site (sexes combined) were (Fig. 5): BDT: W = 0.000008 L2.906, R2 = 0.87, n = 8,818;

SJH: W = 0.00002 L2.734, R2 = 0.87, n = 2,207;

KLH: W = 0.0000001 L3.775, R2 = 0.89, n = 2,560.

Figure 5 Total length–weight relationships of T. swinhonis in (A) Biandantang Lake, (B) Shengjin Lake and (C) Kuilei Lake.

There were significant differences obtained from the statistical comparisons of total length and weight relationships among different sites (for BDT and SJH, F = 58,960, df1 = 1, df2 = 8,816, P ≈ 0 (< 2.2 × 10−16), E = 0.87; for BDT and KLH, F = 15,800, df1 = 1, df2 = 2,205, P ≈ 0 (< 2.2 × 10−16), E = 0.87 and for SJH and KLH, F = 19,870, df1 = 1, df2 = 2,558, P ≈ 0 (< 2.2 × 10−16), E = 0.89). All slope coefficients (b values) were significantly different from the value of 3 for all three lakes (for BDT, t = − 7.84, df = 8,816, P = 5.15 × 10−15, E = 0.08; for SJH, t = −12.22, df = 2,205, P = 2.74 × 10−33, E = 0.26 and for KLH, t = 28.93, df = 2,558, P = 1.74 × 10−159, E = 0.57), indicating an allometric growth pattern of T. swinhonis for all sampling populations.

Growth parameters

Since the age structure of T. swinhonis was simple, we estimated growth parameters using the methods of Powell and Wetherall, which are contained in the ELEFAN I module of the FiSAT II program. The results were BDT: L∞ = 173.25 mm, K = 1.20, t0 = – 1.09, Rn = 0.21; SJH: L∞ = 162.75 mm, K = 1.20, t0 = – 1.08, Rn = 0.29; KLH: L∞ = 215.25 mm, K = 1.20, t0 = – 1.12, Rn = 0.24. The growth equations obtained above showed that T. swinhonis grew rapidly during the first year of it’s life history, obtaining an average L of approximately 159.14, 149.34 and 198.34 mm with the increase of longitude during the first year, respectively, for BDT, SJH and KLH (more than 90% of L∞ at all sites). Annual growth in L decreased sharply after this period and became relatively constant thereafter (Fig. 6). In addition, the GPI (φ′) of T. swinhonis in BDT, SJH and KLH were 4.56, 4.52 and 4.75, respectively.

Figure 6 The von Bertalanffy growth curves of T. swinhonisin in (A) Biandantang Lake, (B) Shengjin Lake and (C) Kuilei Lake as superimposed on the length–frequency histograms.

The different solid lines (blue) corresponded with different cohorts.

Mortality and exploitation rates

The total rates of mortality (Z) obtained through the length-converted catch curves were 5.82 year−1 (with 95% confidence interval of Z: 3.29–8.36 year−1) for BDT, 5.50 year−1 (with 95% confidence interval of Z: 3.31–7.70 year−1) for SJH and 4.55 year−1 (with 95% confidence interval of Z: 3.86–5.23 year−1) for KLH (Fig. 7). The parameters of natural mortality rate (M) estimated by Pauly’s (1980) empirical equation were 1.89 year−1 (BDT), 1.87 year−1 (SJH) and 1.75 year−1 (KLH), respectively. For BDT, SJH and KLH, the values of M/K ratio were 1.58, 1.56 and 1.46 and the fishing mortality rates (F) were 3.93, 3.63 and 2.80 year−1, respectively. The exploitation rates (E) for three sampling populations of T. swinhonis were calculated as 0.68 (BDT), 0.66 (SJH) and 0.62 (KLH) year−1, respectively, which indicated that the stocks in the present study areas have been overfished slightly.

Figure 7 Length-converted catch curves used to estimate total mortality (Z) of T. swinhonis collected from (A) BDT, (B) SJH and (C) KLH along the middle and lower reaches of the Yangtze River basin, China.

Recruitment time and pattern

Recruitment time and pattern of T. swinhonis were similar at all sampling sites. The recruitment pattern was continuous and showed two major peaks throughout the year (Fig. 8), which indicated batch spawning by T. swinhonis. Recruitment occurred from March to September with peaks in April and August. These two peaks accounted for 92.21% (BDT), 88.21% (SJH) and 88.73% (KLH) of total recruitment, respectively.

Figure 8 Annual relative recruitment patterns of T. swinhonis in (A) BDT, (B) SJH and (C) KLH.

Discussion

Size and distribution of length

Knowledge about size structure would be used for understanding age, growth, recruitment and dynamics of the population, which varies with the size structure of the population (Mirzaei, Yasin & Hwai, 2014). Generally, the size structure of one fish species can be represented by the length–frequency distribution (Taiwo, 2010). In the present study, the L of T. swinhonis in BDT ranged from 53.32 to 165.16 mm (the majority in 60–140 mm), in SJH from 51.39 to 154.35 mm (the majority in 80–140 mm) and in KLH from 61.94 to 198.85 mm (the majority in 100–180 mm), and represented an increasing trend in L with increasing longitude. The Kolmogorov-Smirnov tests indicated that length frequencies for T. swinhonis were non-normally distributed at all sites and that there were statistical differences between all sites. Besides, the results obtained from the present study were much larger than the findings of all previous works (Table 2). The disparity observed in the size of T. swinhonis in different studies might be caused by one or more of the following elements: (1) differences of habitat (such as the temperature, density of food and food availability); (2) sample size differences; (3) differences in gear selectivities between studies and (4) differences in the genetic diversity of different geographical populations.

Table 2 Summary of parameters of length–weight relationships of T. swinhonis.

Study area (source)	Total length (cm)	Parameters	n	
Minimum	Maximum	a	b	r2	
Poyang Lake, China (Anonymous, 1974)bc*	5.3	11.1	NA	NA	NA	10	
Dongting Lake, China (Anonymous, 1980)bc*	8.7	12.3	NA	NA	NA	10	
Liangzi Lake, China (Yang, 1986)bc*	5.3	10.5	NA	NA	NA	10	
Bashan and Zhangze reservoir, China (Li et al., 1988)*	NA	13.4	NA	NA	NA	135	
Hunan, hubei, hebei and sichuan China (Chen et al., 1998)bc*	7.2	11.8	NA	NA	NA	23	
Biandantang Lake, China (Zhang, 2005)d*	3.2	9.5	0.0098	3.037	0.98	61	
Niushan Lake, China (Feng et al., 2006)*	2.4	9.5	NA	NA	NA	416	
Niushan Lake, China (Ye et al., 2007)*	5.3	12.1	0.0061	2.845	0.897	289	
The estuary of the Yangtze River, China (Ge, Zhong & Tang, 2009)b*	0.4	2.5	NA	NA	NA	91,849	
Niushan Lake, China (Ye et al., 2012)*	8.4	9.4	NA	NA	NA	2	
Tian-e-zhou Oxbow, China (Wang et al., 2012)bc	5.0	9.3	0.0060	3.220	0.955	103	
Niushan Lake, China (Ye et al., 2014)	4.2	12.8	NA	NA	NA	2,132	
Biandantang Lake, China (the present study)a*	5.3	16.5	0.0061	2.906	0.870	8,818	
Shengjin Lake, China (the present study)a*	5.1	15.4	0.0106	2.734	0.878	2,207	
Kuilei Lake, China (the present study)a*	6.2	19.9	0.0007	3.775	0.886	2,560	
Notes:

n, the number of specimen; a, intercept; b, regression slop; r2, coefficient of determination; NA, not available.

a New records of maximum total length in FishBase.

b Studies with very narrow total length range.

c Small number of specimens.

d Standard length was being used in the study.

* Materials out of FishBase.

Length–weight relationship and growth pattern

The length–weight (L–W) relationship of a fish species always been considered as fundamental to further study, such as fish stock assessment (Afrooz, Gholamreza & Seyed, 2014). Generally, this relationship is used by freshwater ecologists and fisheries managers for (1) predicting the weight (length) for a given length (weight); (2) describing the fitness condition or morphology of one fish species; (3) estimating growth, age structure and many other aspects of a species’ population dynamics and (4) mutual transformation between length and weight for many stock assessment models (Cren, 1951; Ye et al., 2007). Though some fisheries scientists thought it has little value (Hilborn & Walters, 1992), Froese (2006) pointed out that a comprehensive analysis of this relationship of a large number of geographic populations can provide important insights into the biology and ecology of that species.

As previously mentioned, the values of b (slope coefficient) in the L–W relationships in the present study were 2.91 (in BDT), 2.73 (in SJH) and 3.78 (in KLH), respectively. Fish at all three sampling sites displayed allometric growth instead of isometric growth. The results in BDT and SJH were similar with the previous studies in this region. Zhang & Li (2002) reported that the value of allometric parameter was 3.07 in Biandantang Lake, China. Ye et al. (2007) pointed out that this value was 2.85 and presented an isometric growth pattern in Niushan Lake, China. Wang et al. (2012) reported that it was 3.22 in Tian-e-zhou Oxbow, China. The similarity of coefficients between the present study and the previous studies mentioned above may be due to the spatial similarity of the sampling sites in these studies.

However, the slope coefficient (b) estimated by power regression in KLH was 3.78, which was the highest value in all study areas. The observed difference in the value may be attributed to the difference of habitat (especially the observed temperature), the sample size used, diseases and genetic diversity (Wootton, 1998; Ye et al., 2007).

Age structure and growth parameters

Generally, information about age structure and growth parameter of a fish population would be used for fish stock assessment (Hollyman, 2017; Wells et al., 2013), which allows fisheries scientists and managers to understand the dynamics of fish stock and how fish populations respond to the environmental changes. More specifically, it can be an indicator of fish population dynamics and help managers make more informed choices for fish conservation and fisheries management.

In view of this, we first presented the age structure of T. swinhonis in the middle and lower reaches of the Yangtze River basin based on scale readings and Pauly’s empirical equation. These results showed that the age structure of T. swinhonis was very simple and dominant ages were 0+–1+ year old fish (more than 98% at all sites in the present study) which corresponded well with the previous work in the two reservoirs of northern China (Li et al., 1988). The short lifespan of T. swinhonis is very similar with Pseudorasbora parva (Temminck et Schlegel, 1842) and H. leucisculus (two other small dominant species in this region), whose lifespans are no more than 5 and 6 years, respectively (Gozlan et al., 2010; Wang et al., 2013). These two species are two of the most successful invasive species around the world and have been treated as an international pest fish (Gozlan et al., 2010; Patimar, Abdoli & Kiabi, 2008). The simple age structure of these two species was usually considered as one of the major reasons the species became dominant within a very short time in many new regions (Gozlan et al., 2010; Wang et al., 2013). This reason also seems likely for T. swinhonis in the lakes along the middle and lower reaches of the Yangtze River basin.

In terms of growth, the results of VBGMs from the present study indicated that there were some differences in the growth curves of T. swinhonis in different lakes and these differences are mainly reflected in the average asymptotic total length (L∞) as the growth constants (K) were the same (1.20 year−1) in all the three lakes. Moreover, the growth parameters (L∞, K and φ′) obtained by the current study were much larger than the values reported by Ye et al. (2014) in Niushan Lake, China (L∞ = 145 mm, K = 0.66 year−1, φ′ = 4.14) (Table 3). Many studies have documented differences in growth of a given species in different geographical areas (SternPirlot & Wolff, 2006; Vakily, 1992), and the differences might be due to the heterogeneity of habitat and diversity of genetics (Parra et al., 2009; Pauly, 1986; Wootton, 1998).

Table 3 Summary of population characteristics of T. swinhonis in different geographical areas.

Study area	Aging structure	L∞ (mm)	K (year−1)	t0 (year)	ϕ′	Age range (year)	Rn	n	Source	
Bashan reservoir, Chinaa	Scales	NA	NA	NA	NA	1–6	NA	134	Li et al. (1988)	
Zhangze reservoir, Chinaa	
Niushan Lake, China	NA	145.00	0.66	0.30	4.14	NA	0.38	2,132	Ye et al. (2014)	
Biandantang Lake, China	Scales	173.25	1.20	−1.09	4.56	1–4	0.21	8,818	The present study	
Shengjin Lake, China	Scales	162.75	1.20	−1.08	4.52	1–3	0.29	2,207	The present study	
Kuilei Lake, China	Scales	215.25	1.20	−1.12	4.75	1–3	0.24	2,560	The present study	
Notes:

NA, not available; Rn, the goodness of fit index.

a Small number of specimens.

Mortality and fisheries status

Mortality rates are a measure of describing the rate at which fish disappear from a population and are critical parameters in formulating sustainable fishing regulations (Ogle, 2016). In view of this, we estimated the total mortality rates (Z), natural mortality rates (M) and fishing mortality rates (F) of T. swinhonis in the three shallow lakes along the middle and lower reaches of the Yangtze River basin. We present the first evaluation of exploitation rate (E).

Catch curve analysis showed that the values of Z for T. swinhonis were 5.82 (in BDT), 5.50 (in SJH) and 4.55 (in KLH) year−1, indicating a decreasing trend with increasing longitude. We noted that the values of Z estimated in the present study were much bigger than Ye et al. (2014) in Niushan lake, China (Z = 2.92 year−1). The differences of functions of these lakes and the fishing effort may explain these differences (Table 4).

Table 4 Estimated mortality rates of T. swinhonis derived from different areas.

Study areas (Source)	Z (year−1)	M (year−1)	F (year−1)	
Biandantang Lake, China (the present study)ad	5.82	1.89	3.93	
Shengjin Lake, China (the present study)ad	5.50	1.87	3.63	
Kuilei Lake, China (the present study)b	4.55	1.75	2.80	
Niushan Lake, China (Ye et al., 2014)c	2.92	1.35	1.57	
Notes:

a National Wetland Park.

b Wellhead protection zone.

c Aquaculture base.

d Having fishman.

Values for M were similar between lakes: BDT = 1.89, SJH = 1.87 and KLH = 1.75. These values were slightly higher than the findings of Ye et al. (2014) in Niushan Lake, China (M = 1.35). Meanwhile, the values of M/K ratio in our study were 1.58 (in BDT), 1.56 (in SJH) and 1.46 (in KLH), within the interval of 1.12–2.50 for most fish (Pauly, 1980), indicating our estimate of M is reasonable. However, Pauly (1980) also postulated that M should be in the range of 0.2–0.3, indicating T. swinhonis in the region of the Yangtze River basin was characterized with relatively high M, not uncommon for short-lived species.

The fishing mortality rate (F) in BDT, SJH and KLH were 3.93, 3.63 and 2.80 year−1, values that were much higher than those found by Ye et al. (2014) in Niushan Lake, China (F = 1.57). High values of F at all sites were likely related to the morphological characteristics (barbs on the dorsal fin of T. swinhonis made them more susceptible to capture by multi-gillnets than other species). Certainly, the difference between the present study and previous work conducted by Ye et al. (2014) may be explained by the gear employed. We used the float multi-gillnets in the present study which targeted the preferred water layer (pelagic) of T. swinhonis while sink multi-gillnets was used by Ye et al. (2014) in Niushan Lake, China targeted the bottom.

Based on estimates of F and Z, the exploitation rates (E) in BDT, SJH and KLH were 0.68, 0.66 and 0.62, respectively. According to the rule of thumb (when E was more than 0.5, the fish stock was undergoing overfishing) proposed by Gulland (1965), we found that the stocks in the present study areas had been overfished slightly, which should be considered by fisheries managers in the future.

Recruitment time and pattern

Recruitment is a very important period in a fish’s life history and can be expressed as the number of fish that reach a certain age or length, which can give us an insight into the future viability of a fish population or predict the possible harvest from the population (Ogle, 2016).

The spawning time and pattern of T. swinhonis in the present study was identical at all sites and annual recruitment consisted of two seasonal pulses (batch spawning) which occurred between months of March to September. Long recruitment time and strong reproductive ability (L. S. Liu, 2017, unpublished data) may be the main reasons why the species became dominant in these lakes. Despite close agreement with annual changes of gonadosomatic index and distributions of egg size (L. S. Liu, 2017, unpublished data), the results in the present study are not consistent with Ye et al. (2014), who found the annual recruitment only consisted of one plus (single spawning) which occurred between April and August. This is an important topic for future research.

Conclusions

This study is the first attempt to elucidate the mechanism of population outbreaks of T. swinhonis in the lakes along the middle and lower reaches of the Yangtze River basin based on novel and comprehensive biological data. Current results showed that, like many other freshwater fishes, brief generation-time, fast growth rate, relatively high natural mortality rates and strong reproductive capacity (r-strategies) may be the main causes that this fish became the most dominant fish species in many lakes of China. Thus, to control the population of T. swinhonis, catching it moderately before its spawning season or discreetly releasing predator fish is proposed (Ye et al., 2006). Certainly, more field investigations should be conducted on this species in the future, especially on the reproductive biology and population dynamics of this species.

Supplemental Information

Supplemental Information 1 Raw datasets and code for the present study.

Original dataset and code using in this study.

Click here for additional data file.

We are grateful to Ms. Jin Yuan, Mr. Mantang Xiong, Mr. Zhan Mai, Mr. Lisheng Liu and Mr. Xinnian Chen for their assistance in the field sampling. We would also like to give our sincere appreciations to two anonymous referees for their valuable comments and suggestions, leading the huge improvement of the earlier manuscript of this paper.

Additional Information and Declarations

Competing Interests

Author Contributions

Animal Ethics

Field Study Permissions

Data Availability

The authors declare that they have no compting interests.

Xianghong Dong conceived and designed the experiments, performed the experiments, analyzed the data, contributed reagents/materials/analysis tools, prepared figures and/or tables, authored or reviewed drafts of the paper, approved the final draft.

Tao Xiang performed the experiments, contributed reagents/materials/analysis tools, prepared figures and/or tables, authored or reviewed drafts of the paper.

Tao Ju performed the experiments, contributed reagents/materials/analysis tools, prepared figures and/or tables, authored or reviewed drafts of the paper.

Ruojing Li performed the experiments, contributed reagents/materials/analysis tools, authored or reviewed drafts of the paper.

Shaowen Ye performed the experiments.

Sovan Lek analyzed the data, prepared figures and/or tables, approved the final draft.

Jiashou Liu conceived and designed the experiments, authored or reviewed drafts of the paper, approved the final draft.

Gaël Grenouillet authored or reviewed drafts of the paper, approved the final draft.

The following information was supplied relating to ethical approvals (i.e., approving body and any reference numbers):

The ethics committee of the Institute of Hydrobiology, Chinese Academy of Sciences provided full approval for this purely research (Y216011101).

The following information was supplied relating to field study approvals (i.e., approving body and any reference numbers):

The ethics committee of the Institute of Hydrobiology, Chinese Academy of Sciences provided the field permit (Y216011101).

The following information was supplied regarding data availability:

The raw datasets and the code are available in the Supplemental File.

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
