# Peer review of "Age, growth, mortality and recruitment of thin sharpbelly Toxabramis swinhonis Günther, 1873 in three shallow lakes along the middle and lower reaches of the Yangtze River basin, China"

_PeerJ, doi:10.7717/peerj.6772_

## Round 0.1 · original submission · Major Revisions

Dear Xianghong and Jiashou,

Thank you for your submission to PeerJ. Your article has been reviewed by 2 expert reviewers who have made several comments. Both have recommended Major Revisions, and Reviewer 1 has provided an additional document for you. Please address their comments in detail

Reviewer 1 ·

Basic reporting

The article is fairly well written, far better than i could do were i to submit an article to a Chinese fisheries journal! That being said there are plenty of cases where the grammar could be improved. I have offered numerous suggested rewrites in my attached comments.

There are quite a few instances where the literature cited and the citations in the text do not match up. Often it seems to be a misstated year or a missing authors name on a multi-author citation, but there are enough of these that it poses an impediment to approval of the manuscript for publication. I have identified all of these in the line-by-line comments attached.

Tables and Figures are adequate. I believe Tables 1 and 2 are misordered in the text, reference to Table 1 really refers to Table 2. This needs to be clarified and fixed.
Fig. 5 needs clarification as to where all the lines on the graphs come from, if they are von bert curves, why are there more than one per lake? This could be a function of the software package that generates them, but i am not aware, and i suspect others without a lot of experience with the ELEFAN software module will not know what these are either.

Experimental design

The research question was wide-casting, to study and provide basic life history information on an understudied species. This is the type of information that is often overlooked by managers looking only to mange the important high profile species. Methods were usual and appropriate and described in sufficient detail.

Validity of the findings

The study presents valid results for a previously un- or understudies species. Methods used were de riguer for the field and well explained. My main questions concern the use of the tests for normality of the data. The length data for all three lakes were found to be non-normal, or abnormal in their words. But no explanation was then given of what that meant in the context of the study. What should then have been done with the length data if it were deemed non-normal? is there a transformation it needs to go under? does the finding of non normality dictate then the usage of specified statistical tests? i think in this light, a better explanation of the assumptions of the tests used might be justified.

The data was collected in an efficient manner, lab methods (reading of scales) was routine. Conclusions were brief and i think could be expanded upon. AS i stated in the comments, there could be much more expanded discussion on what might have led to differences in some of the life history characteristics found between lakes, for example the very different slope coefficient for the one lake indicating much more robust fish. Some expanded discussion on why is warranted. Also more general discussion about differences, if any, in the habitat of the three lakes. As i mention in a comment, looking at Fig. 1 shows that one of the lakes has much more shoreline than the other two (finger development). This could be crucial to things like available nursery habitat, etc, and yet is not discussed.

Additional comments

This paper is a good attempt to describe differences in age growth and mortality traits of a very short lived important species in the riverine systems of China. More care needs to be paid to the literature cited section, there are several mis-citations. There is some confusion in the text as to the citing in the proper order of mention of Tables 1 and 2. Some clarification is needed on the components of the box plot, maybe in the legend, what is mean, what is observed points, etc. The figure for the von bertalanffy curves over the LF histograms needs clarification as to what all the lines are, my usual contact with von bert curves is one line per population, why multiple lines?
Your putting this manuscript into the English language was very good, but as expected there are places where there are minor mistakes, i made as many changes as i could in my comments to assist you in making it a better manuscript.
I would like to see you expand the introduction to elaborate more if you can on the importance of this species. Is it important simply as forage for other carnivores, or is it utilized by man as food? Are there currently regulations on the catch of this species? is it even the target of a fishery? You mention it accounts for 85% of catch in some lakes, but who is catching it? targeted commercial fishing? subsistence artisanal private fishermen? is there a market for it?

I think you need to revisit the discussion of longitudinal axis, as i outlined in my line by line comments. and also, while it is true that the lakes all seem to be marginally separated by longitudes, looks like a 10 degree wide difference for all three lakes, why is longitude relevant? usually it seems its a latitudinal difference (north to south) that matters more, and these are basically all at the same latitude? what is it about this region of China where longitudinal separation might be important?

I would like to see you equally expand the discussion as to the importance of your findings and how you envision they will be used by fishery managers in China? These are all points that the readers would need to assess the relevancy of this work.

Annotated reviews are not available for download in order to protect the identity of reviewers who chose to remain anonymous.

Reviewer 2 ·

Basic reporting

While the structure of the article is well-organised, there are many minor grammatical and phrasing errors, as well as numerous colloquialisms (e.g. referring to the focal species as a ‘trash’ species). These are too numerous to list, so I leave the task to the editorial team.

Experimental design

The second aim (b) of this paper needs to be rephrased to avoid suggestion that the cause of large population size for the study species can solely be determined by investigating demography. I understand what the authors are trying to say, but in its current form, the aim cannot be achieved because there are many other potential causes for large population size, for example, competitive or predatory release. I suggest rewording the aim to state that the study ‘aims to identify demographic characteristics consistent with rapid population expansion’. I also recommend that background information on the link between particular life-history strategies and population size be added to the Introduction, so the reader understands what the study is trying to achieve in this regard.

The use of scales for ageing (lines 147) instead of otoliths requires justification. Scales have generally proven less reliable ageing structures than otoliths, because they tend to underestimate ages in older individuals. While this species appears short-lived, and may be a viable candidate for ageing via scales, this should be verified, or at the very least, discussed and qualified.

Related to the previous issue is the lack of age validation in the current study. The annual periodicity of marks counted within the scales must be verified before absolute ages can be interpreted with confidence. This step is essential in the current study because it presents the first age data for the species. The use of Pauly’s maximum age formula (line 154) is not sufficient for age validation, because the value for K used in that formula relies on the age data from the study that was entered into the VBGM (ie the t-max estimate is not independent). I strongly recommend conducting Marginal Increment Analysis (MIA) to demonstrate a single peak in the marginal increment width (distance between the last complete increment and the scale edge along the counting transect) over a 12 month period. The authors have a sufficient collection of individuals for this analysis at an appropriate temporal resolution (monthly collection). Without confidence that marks within the scales were formed annually, few of the conclusions within the paper can be supported. I also recommend the authors provide a figure showing some images of their scales with the annual increments marked. This will hopefully increase the readers faith in the scale-reading technique.

Validity of the findings

Lines 283-285: it’s unclear to me why the authors need to justify the exponent of the LW relationships using a range of known values across previously studies species. The only reasons I can think of might be that the authors believe that sampled lengths and weights are not representative of the populations, e.g. the passive sampling achieved using gill nets precluded capture of less active fish that might have lighter weights for a given length, or the length and weight measurements in the current study were inaccurate. If these are not the case, I suggest removing the comparison to other species for validation purposes. Of course, comparison with other species might still be valid in the discussion for some biological purpose.

Lines 298-299: this is certainly not the case. While age and growth information are highly desirable, there are many rudimentary stock assessment methods that do not require this information. I recommend the authors state the assessment methods that such information would be useful for, rather than engage in sweeping statements of importance (which seem common for numerous variables within the Discussion). I’m also not clear why the authors mention stock assessment for the study species, because it was made clear in the Introduction that they are of little commercial interest (bycatch species).

---

## Round 0.2 · Minor Revisions

This revised manuscript has been improved over the first version. However, there are still some minor corrections. Please follow the comments from the reviewer. After this revision, this manuscript can be accepted for a publication.

Reviewer 1 ·

Basic reporting

Basic reporting is improved over the first version i reviewed. the authors have incorporated many changes that I and other reviewers, i suspect, recommended. However, there is still room for improvement. I have attached an annotated scanned pdf with further suggestions for improvement and editorial improvement (rephrasing, etc) that i believe would make the paper more readable.

Experimental design

The experimental design of this study is acceptable, it was a field study collecting specimens for fishery research, and was conducted in a standard manner. Laboratory processing and analyses of data to arrive at fishery products (life history parameters) was all conducted using appropriate methods.

Validity of the findings

Findings are valid. There are areas where further discussion would make this a much better paper. The answers may not be known, but the authors could lead the discussion to invite the readers to ponder why certain environmental variables might be responsible for certain results, say (e.g., habitat differences in lakes and why one lake might have substantially different growth parameters coming out of it, etc.) I have identified a couple of places in the manuscript where i think expanded discussion is warranted.

One of these areas where more discussion is needed is the result of the insertion of a single sentence at the end of the discussion that was not in the first version, a sentence where you talk about moderate fishing or introducing discretely a control predator for population control....wow! this came out of nowhere, and changes the whole context that i viewed this study from.....i thought the general direction of the paper was that the species was of moderate importance from both a fishery standpoint and an ecological forage component standpoint.....but this sentence changes all that......because of this sentence you need to either embellish the introduction to cover this, or expand the discussion greatly.....

Additional comments

This paper is much improved, but still needs what i would call minor revisions. You can see many of my suggestions on the attached scanned copy, i hit my major salient points above....two smaller points that you may want to address:

the interchanging use of TL and L for length....chose one and stay with it....

the use of 0.01 mm....this is an incredibly small resolution for a fish length, and i would argue biologically meaningless for a growth study on fishes....if we were studying dinoflagellates, sure....the use of 0.01g for gonad weight is fine, that is the norm, but i think lengths can reasonably be in whole mm...

Annotated reviews are not available for download in order to protect the identity of reviewers who chose to remain anonymous.

---

## Round 0.3 · accepted · Accept

This manuscript can be accepted and published.

#